# Impact of Empirical Antimicrobial Treatment on Patients with Ventilator-Associated Pneumonia Due to *Stenotrophomonas maltophilia*

**DOI:** 10.3390/antibiotics13080729

**Published:** 2024-08-03

**Authors:** Pirawan Khunkit, Pisud Siripaitoon, Yongyut Lertsrisatit, Dissaya Watthanapaisal, Narongdet Kositpantawong, Siripen Kanchanasuwan, Nadia Cheh-oh, Sorawit Chittrakarn, Tanapat Jaroenmark, Natnicha Poonchuay, Sarunyou Chusri

**Affiliations:** 1Department of Pharmaceutical Care, School of Pharmacy, Walailak University, Tha Sala, Nakhon Si Thammarat 80160, Thailand; pirawan.kh@wu.ac.th; 2Department of Internal Medicine, Faculty of Medicine, Prince of Songkla University, Songkhla 90110, Thailand; grippen45@gmail.com (P.S.); poom_032@yahoo.com (N.K.); kaymed29@yahoo.com (S.K.); sorawit.c@psu.ac.th (S.C.); tanapat.jaroenmark@gmail.com (T.J.); 3Department of Clinical Pharmacy, Faculty of Pharmaceutical Sciences, Prince of Songkla University, Songkhla 90110, Thailand; yongyut@pharmacy.psu.ac.th (Y.L.); dissaya@pharmacy.psu.ac.th (D.W.); nadia.ch@psu.ac.th (N.C.-o.); 4Drug and Cosmetics Excellence Center, Walailak University, Tha Sala, Nakhon Si Thammarat 80160, Thailand; 5Department of Biomedical Sciences, Faculty of Medicine, Prince of Songkla University, Songkhla 90110, Thailand

**Keywords:** antimicrobial treatment, ventilator-associated pneumonia, *S. maltophilia*

## Abstract

This retrospective study was conducted to evaluate the characteristics and outcomes of patients with ventilator-associated pneumonia (VAP) caused by *Stenotrophomonas maltophilia* (*S. maltophilia*), focusing on the impact of appropriate empirical antimicrobial treatment. Of the enrolled 240 patients with VAP due to *S. maltophilia* (median age: 45 years) in a tertiary-care hospital in southern Thailand between January 2010 and December 2021, 90% had medical comorbidities and 91% had previously received carbapenems. In addition, only 45% of the patients were initially admitted to the intensive care unit. Patients administered appropriate empirical antimicrobial treatment including colistin alone and colistin plus TMP-SMX or fluoroquinolone-based regimens had significantly lower 14-day, 30-day, and in-hospital mortalities, compared with those who did not receive appropriate empirical antimicrobial treatment (21% and 2% vs. 31%; 30% and 5% vs. 44%; and 30% and 12% vs. 53%, respectively). Thus, the use of appropriate empirical antimicrobial treatments led to a significantly reduced length of hospital stay, duration of ventilation, and hospital costs. The current study suggests that the use of appropriate empirical antimicrobial treatment based on susceptibility testing without considering pharmacokinetic properties and administration dosages improves the outcomes of patients with VAP due to *S. maltophilia*.

## 1. Introduction

*Stenotrophomonas maltophilia* (*S. maltophilia*) has emerged as an important pathogen responsible for hospital-associated infections, particularly in patients with debilitating medical conditions [1]. The most common clinical manifestation of this pathogen is pneumonia, particularly ventilator-associated pneumonia (VAP) [2]. This pathogen causes VAP through its ability to colonize respiratory tract epithelial cells and surfaces of medical devices [3]. This ability is based on the properties of its flagella and fimbrial adhesins, biofilm formation, and resistance to complement-mediated cell killing [4,5,6,7]. Thus, this organism has been found as a contaminant in hospital environments, particularly in several kinds of medical devices, specimen collection tubes, chlorhexidine-based disinfectants, drinking water supply, and chlorinated sterile water [3]. In addition, *S. maltophilia* exhibits intrinsic resistance to numerous antimicrobial agents with intrinsic mechanisms including multidrug-efflux pumps and low permeability as well as the acquisition of resistance through the uptake of resistance genes located on integrons, transposons, and plasmids which are responsible for β-lactamases, aminoglycoside-modifying enzymes, and alteration of folate synthesis [1,3]. Infections due to *S. maltophilia* have raised concerns over the number of surveillance cases, particularly in European and Southeast Asian and Pacific countries [1,3,8,9]. These infections commonly emerged among vulnerable patients with severely compromised health status, retaining medical devices, exposure to broad-spectrum antimicrobials, and long hospital stays [1,3]. The occurrence of outbreaks with the local spread of these infections was frequently noticed, and global spread of specific strains was identified [2,3,8,9].

Although *S. maltophilia* is classified as a non-inherently virulent organism, VAP caused by this organism has a high mortality rate owing to the limitation of appropriate antimicrobial treatment options [1,8,9]. This can be explained as follows: First, carbapenems, often used as empirical treatment for hospital-acquired infections, have unfavorable antimicrobial susceptibility due to the intrinsic resistance of *S. maltophilia* [9]. Second, although *S. maltophilia* is susceptible to colistin, which is commonly prescribed empirically for suspected carbapenem-resistant Gram-negative bacterial infections, the pulmonary tissue penetration of this agent is relatively poor [10,11]. Third, even though co-trimoxazole (trimethoprim-sulfamethoxazole [TMP-SMX]) and fluoroquinolones have high susceptibility against *S. maltophilia*, these antimicrobial agents are not commonly prescribed for hospital-acquired infections due to poor susceptibility against carbapenem-resistant Enterobacterales (CRE), carbapenem-resistant *Acinetobacter baumannii* (CRAB), and difficult-to-treat resistant *Pseudomonas aeruginosa* (DTRPA) [12]. Then, treatment of VAP caused by *S. maltophilia* is extremely hampered because this organism is intrinsically resistant to several kinds of antimicrobial agents; thus, the existing potential antimicrobial agents against this organism have unclear vitro susceptibility testing [8,9].

Timely and appropriate empirical antimicrobial therapy is associated with reduced mortality in patients with VAP [10]. However, several challenges have been reported regarding the use of empirical antimicrobial treatment options for VAP caused by *S. maltophilia* [8,9,13,14,15]. In this observational study, we aimed to characterize patients with VAP caused by *S. maltophilia*, focusing on appropriate empirical antimicrobial agents, to reduce mortality rates and hospital expenditure.

## 2. Materials and Methods

### 2.1. Patients and Setting

This retrospective study was conducted among adult patients (aged ≥ 18 years) diagnosed with VAP caused by *S. maltophilia* admitted to Songklanagarind Hospital, a tertiary care hospital and referral center in southern Thailand between January 2010 and December 2021. These patients were diagnosed based on the presence of a new and persistent infiltrate on chest radiographs plus two or more of the three following criteria: fever > 38.3 °C, leukocytosis > 12 × 10^9^/mL, with or without purulent tracheobronchial secretions after 48 h of intubation, and the need for mechanical ventilation [16]. To eliminate duplications, only data from the first episode and complete medical records were included in the analysis. Patient clinical data were reviewed from electronic medical records, and microbiological data were extracted from the hospital microbiology database.

### 2.2. Bacterial Identification and Antimicrobial Susceptibility

All clinical isolates were identified as *S. maltophilia* by conventional biochemical tests, followed by matrix-assisted laser desorption/ionization time-of-flight mass spectrometry methods. Antibiotic susceptibility testing was initially performed using clear zone diameters determined using the Kirby–Bauer disk diffusion method. The minimal inhibitory concentrations of all isolates were determined using conventional microdilution techniques. Antimicrobial susceptibility was determined according to the Clinical and Laboratory Standards Institute [17].

### 2.3. Study Design and Data Collection

All relevant data including demographic and clinical variables, such as age, sex, body mass index, laboratory results, and presence of underlying conditions (cerebrovascular disease, cardiovascular disease, chronic kidney disease, and diabetes mellitus) were obtained from electronic medical records. Immunocompromised status was defined as having an absolute neutrophil count < 0.5 × 10^9^/L for >2 weeks or being on immunosuppressive therapy (chemotherapy for 6 weeks, corticosteroids at a dosage >15 mg of prednisolone daily for >2 weeks, or disease-modifying antirheumatic drugs for 4 weeks) [15]. The degree of severity of the disease was determined based on instances of VAP, initial ICU admission, patient’s acute physiology and chronic health evaluation (APACHE) II score [18], presence of bacteremia, use of invasive medical devices, radiography data, and microbiology data. Treatment data included the appropriateness of empirical antibiotic therapy and duration of antibiotic treatment. The study design was based on a comparison of outcomes among three groups of patients with VAP caused by *S. maltophilia*. The first group included patients who did not receive any antimicrobial agents effective against *S. maltophilia* as empirical regimens. The second group included patients who received colistin alone as the appropriate empirical regimen. The third group included patients who received colistin and TMP-SMX or ciprofloxacin/levofloxacin as an appropriate empirical regimen. Appropriate empirical antimicrobial agents were determined using only susceptibility testing information without concern for pharmacokinetic properties and administration dosages. The clinical outcomes of this study included 14-day mortality, 30-day mortality, in-hospital mortality, and length of hospital stay, as well as hospital expenses, which were divided into antimicrobial treatment costs and other hospitalization-associated (non-antimicrobial) costs. Risk factors associated with mortality were analyzed based on potential variables extracted from electronic medical records.

### 2.4. Statistical Analysis

The characteristics and clinical outcomes of the patients were compared by tabulation, followed by the chi-square or Fisher’s exact test as appropriate for categorical variables and Student’s *t*-test for continuous variables. Differences in variables were expressed as odds ratios and 95% CI. Variables with *p*-values < 0.2 and variables with potential clinical relevance were included in multivariate logistic regression models. These models were fitted to determine the effect of each variable and expressed as adjusted ORs. The final model included all independent variables. The significance level was set at *p* < 0.05. Associations between variables and outcomes were expressed as adjusted ORs and 95% Cis. Survival analysis using Cox proportional hazard regression was performed to assess differences in survival duration after receiving empirical antimicrobial agents. Start time was defined as the day on which the empirical antimicrobial agents were administered, while end time was defined as the date on which the patient outcomes were documented. The influence of each relevant variable was expressed as a hazard ratio (HR) and 95% CI.

## 3. Results

Between January 2010 and December 2021, 261 patients were diagnosed with VAP caused by *S. maltophilia*, of whom 240 patients were enrolled in this analysis. The study enrollment process is shown in Figure 1. The results of antimicrobial susceptibility testing of *S. maltophilia* isolates from the 240 patients are shown in Table 1. The highest susceptibility rates were observed for TMP-SMX (97%), tigecycline (96%), levofloxacin (95%), and ciprofloxacin (95%), while the rates of susceptibility to colistin and ceftazidime were relatively low at 77% and 44%, respectively.

Table 2 shows the baseline demographic characteristics of the 240 patients. Most of them (90%) had medical comorbidities, with a median Charlson comorbidity index of 6 (95% confidence interval [CI] 5–8). In addition, most of them (91%) had received carbapenems prior to the development of VAP due to S. maltophilia. Less than half of the patients (45%) were initially admitted to the intensive care unit (ICU), and the median initial acute physiology and chronic health evaluation (APACHE) II score was 17 (95% CI 13–21). The most common empirical treatment regimen was carbapenem (98%), followed by colistin (30%), TMP-SMX (13%), and fluoroquinolone (8%). Less than half of the patients (44%) received appropriate empirical antibiotics, including colistin monotherapy-based regimen (26%) and colistin plus TMP-SMX or a fluoroquinolone-based regimen (18%) None of the patients with VAP due to colistin-resistant *S. maltophilia* received colistin plus TMP-SMX or a fluoroquinolone-based regimen.

Table 3 shows the results of the comparison of characteristics and outcomes of 135 patients who did not receive appropriate empirical antibiotics, 63 who received appropriate empirical antimicrobial treatment with a colistin monotherapy-based regimen, and 42 who received an appropriate empirical antimicrobial treatment with colistin plus TMP-SMX or a fluoroquinolone-based regimen. The number of comorbidities, rate of initial ICU admission, and APACHE II scores among those who received appropriate empirical antimicrobial treatment were relatively high, compared with those who did not receive appropriate empirical antibiotics. The 14-day mortality of the patients who did not receive appropriate empirical antimicrobial agents (31%) were significantly different from those who received empirical antimicrobial treatment with colistin monotherapy-based regimen (21%) and those who received empirical antimicrobial treatment with colistin plus TMP-SMX or fluoroquinolone-based regimen (2%) (*p*-value < 0.001). The 30-day mortality of the patients who did not receive appropriate empirical antimicrobial agents (44%) were significantly different from those who received empirical antimicrobial treatment with a colistin monotherapy-based regimen (30%) and those who received empirical antimicrobial treatment with colistin plus TMP-SMX or a fluoroquinolone-based regimen (5%) (*p*-value < 0.001). The in-hospital mortality of the patients who did not receive appropriate empirical antimicrobial agents (53%) were significantly different from those who received empirical antimicrobial treatment with colistin monotherapy-based regimen (30%) and those who received empirical antimicrobial treatment with colistin plus TMP-SMX or fluoroquinolone-based regimen (12%) (*p*-value < 0.001). The length of hospital stay after VAP treatment, duration of ventilation since VAP diagnosis, and hospital costs, among those who received appropriate empirical antimicrobial treatment, were significantly favorable, particularly in those who received colistin plus TMP-SMX or a fluoroquinolone-based regimen, compared to those who did not receive appropriate empirical antimicrobial agents. Figure 2 illustrates the Kaplan–Meier curve of in-hospital survival among patients with VAP due to *S. maltophilia* who received appropriate empirical antibiotics, which differed significantly from that of those who did not (*p* < 0.001, log-rank test).

Factors associated with in-hospital mortality in 240 patients with VAP due to *S. maltophilia* are shown in Table 4. Only the initial APACHE II score and use of appropriate empirical antibiotics, including colistin-based regimens and colistin plus TMP-SMX or fluoroquinolone-based regimens, were associated with in-hospital mortality with an adjusted OR (95% CI) of 1.32 (1.05, 2.43), 0.25 (0.12, 0.53), and 0.08 (0.02, 0.19), respectively. The survival analysis with the Cox proportional hazard model showed that the factors influencing in-hospital mortality were APACHE II score (HR, 1.12; 95% CI, 1.09 to 1.91; *p* = 0.021), and appropriate empirical antibiotics, including colistin-based regimens (HR, 0.42; 95% CI, 0.13 to 0.75; *p* = 0.002) and colistin plus TMP-SMX or fluoroquinolone-based regimens (HR, 0.21; 95% CI, 0.10 to 0.43; *p* = 0.001).

## 4. Discussion

The current study indicated that VAP due to *S. maltophilia* had high 14-day, 30-day, and in-hospital mortality rates at 31%, 44%, and 53% among the patients who did not receive appropriate empirical antimicrobial treatment. Among the appropriate empirical antibiotic regimens, colistin plus TMP-SMX or fluoroquinolone-based regimen yielded significantly favorable clinical and non-clinical outcomes.

This study showed that *S. maltophilia* was susceptible to TMP-SMX, levofloxacin, ciprofloxacin, and tigecycline. TMP-SMX was suggested to be the most effective antibiotic against VAP caused by *S. maltophilia*, followed by fluoroquinolones. These findings were like those of surveillance studies conducted between 1998 and 2021 [1,2,3,13,14]. Like a previous study conducted among patients with bacteremia due to *S. maltophilia* at the present study location, the rate of resistance to TMP-SMX was relatively low, while the rate of resistance to colistin was high at 35% [15].

Most of the patients in the present study (91%) had previously received carbapenem before developing VAP due to *S. maltophilia*, reflecting problematic concerns with the use of carbapenem in hospitalized patients, which may have led to the emergence of *S. maltophilia* infection due to its intrinsic resistance to carbapenem. This finding was consistent with those of several studies that focused on the risk factors for this infection [2,19,20]. Contrary to previous studies, the present study showed that VAP due to *S. maltophilia* was not restricted to patients admitted to the ICU, as approximately 55% of the patients developed this infection during admission to general wards [2,3,13,14]. Regarding the setting with limited ICU bed capacities, concerns of this infection should be extended beyond patients who have undergone mechanical ventilation not only in the ICU but also in general wards.

The in-hospital mortality rate among the patients who did not receive appropriate empirical antimicrobial agents was relatively high (53%). This finding was like previous studies of VAP due to *S. maltophilia* [2,3,13,14]. In this study, APACHE II, indicating the severity of VAP due to this organism was significantly associated with in-hospital mortality with an OR (95% CI) of 1.32 (1.05,2.43). Like previous studies of VAP due to several Gram-negative bacteria, the initial conditions of the patients with VAP were independent of the appropriateness of empirical antimicrobial regimens [21,22,23]. Resembling previous studies, appropriate empirical antimicrobial regimens for the patients with VAP due to *S. maltophilia* were significantly associated with lower mortality [13,14,21,22,23]. Thus, our study demonstrated the different appropriate empirical regimens yielded different effects on the mortality of the patients. These findings filled the gap of previous studies on VAP due to *S. maltophilia* which demonstrated the benefits of appropriate empirical antimicrobial agents regardless to the type of antimicrobial agents [2,3,19,20].

The current study showed the clinical and non-clinical benefits associated with appropriate antimicrobial treatment. These findings suggest that the use of appropriate empirical antibiotics yielded lower 14-day, 30-day, and in-hospital mortality rates, as well as decreases in the length of hospital stay, duration of ventilation, and hospital costs, respectively. These findings are consistent with those of studies on VAP that focused on empirical antibiotic regimens [21,22,23]. Additionally, this current study showed lower mortality among the patients who received appropriate empirical antibiotics with colistin plus trimethoprim-sulfamethoxazole or fluoroquinolone(s), compared to those who received empirical antibiotics with colistin monotherapy.

Regarding evidence suggesting a correlation between pharmacokinetics (PK) and antibiotic efficacy in patients with VAP, the present study showed different outcomes between two appropriate empirical antibiotic regimens [23,24]. Despite the rate of susceptibility to colistin in this study being acceptable at 77%, the benefits of the drug, including reduced mortality rate, length of hospital stay, duration of ventilation, and hospital cost, seemed less favorable, compared with the benefits of combined colistin and TMP-SMX/fluoroquinolone regimen. This can be explained by the limited ability of the drug to pass through the respiratory epithelium, basement membrane, and vascular endothelium due to its hydrophilic nature and relatively large and positively charged molecules [23,25].

This study has several limitations that should be acknowledged. First, this study was conducted in a single tertiary hospital, and the findings can be generalized only to similar settings. Understanding that the number of patients with VAP due to *S. maltophilia* in other settings is relatively low, this study demonstrated the clinical characteristics of patients with VAP due to *S. maltophilia*, suggesting the use of empirical antimicrobial agents. Second, owing to the retrospective nature of this study, data addressing clinical judgment were lacking. Third, no patient with VAP due to colistin-resistant *S. maltophilia* received an empirical antibiotic regimen of colistin plus TMP-SMX/fluoroquinolone. Data demonstrating the efficacy of TMP-SMX/fluoroquinolone alone as an empirical antibiotic was lacking. Fourth, the data used in this study were retrieved from January 2010 to December 2021, and there are no standard recommendations for the definitive treatment of infections caused by *S. maltophilia*. In total, 28% of the patients received monotherapy with colistin plus TMP-SMX or fluoroquinolone as a definitive treatment, regardless of disease severity. In contrast, recent recommendations suggest the use of combination therapy as a definitive treatment for moderate-to-severe cases [26]. Last, the definition of appropriate empirical antimicrobial agents was determined by only antimicrobial susceptibility testing regardless of pharmacokinetic properties as well as the dosage of antimicrobial agents which had potential impacts on outcomes of the patients.

## 5. Conclusions

VAP due to *S. maltophilia* yields a high mortality rate and excessive hospital expenditure. Appropriate empirical antimicrobial treatments improve clinical outcomes as well as decrease ventilator use, length of hospital stays, and hospital expenditure. A well-designed clinical trial should be conducted to determine the impact of appropriate empirical antibiotic regimens on outcomes for patients with potentially emerging VAP due to *S. maltophilia*.

## Figures and Tables

**Figure 1 antibiotics-13-00729-f001:**
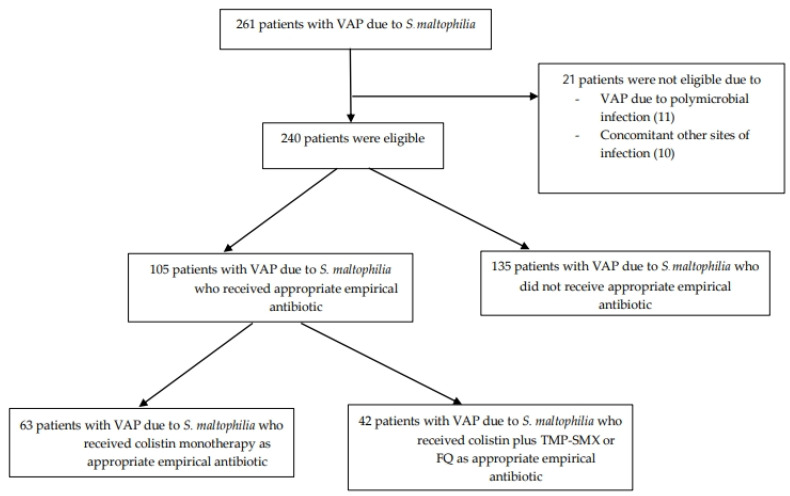
Study enrollment flow chart. VAP: ventilator-associated pneumonia, TMP-SMX: trimethoprim-sulfamethoxazole, FQ: fluoroquinolone.

**Figure 2 antibiotics-13-00729-f002:**
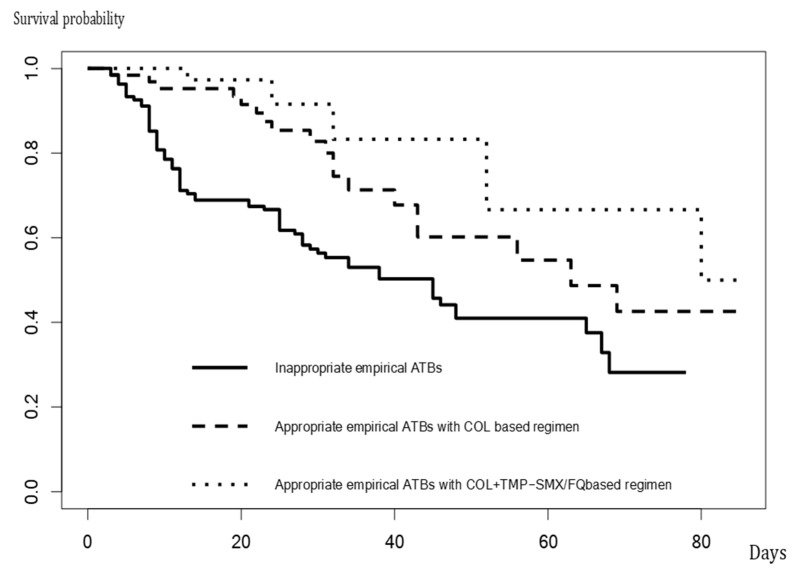
The Kaplan–Meier curve of in-hospital survival among patients with VAP due to *S. maltophilia*.

**Table 1 antibiotics-13-00729-t001:** Antibiotic resistance profiles of 240 *S. maltophilia* isolates obtained from patients with ventilator-associated pneumonia.

Antibiotics	No. of Resistant Isolates (%)
Chloramphenicol	32 (13)
Colistin	55 (23)
Ciprofloxacin	11 (5)
Levofloxacin	12 (5)
Trimethoprim-sulfamethoxazole	7 (3)
Tigecycline	9 (4)
Ceftazidime	135 (56)

**Table 2 antibiotics-13-00729-t002:** Characteristics of 240 patients with VAP due to *S. maltophilia*.

Parameter	Patients with VAP due to *S. maltophilia* (N = 240)
**Demographics**	
Age, median (IQR), [range]	45 (41–74) [19–96]
Male sex, n (%)	158 (66)
Comorbidities, n (%)	217 (90)
Immunocompromised status, n (%)	12 (5)
Obesity, n (%)	176 (73)
Diabetes mellitus, n (%)	82 (34)
Hypertension, n (%)	124 (52)
Chronic kidney disease(s), n (%)	30 (13)
Cardiovascular disease(s), n (%)	48 (20)
Cerebrovascular disease(s), n (%)	8 (3)
Chronic pulmonary disease(s), n (%)	65 (27)
Solid organ malignancy, n (%)	6 (3)
Hematologic malignancy, n (%)	49 (2)
Charlson comorbidity index, median (IQR), [range]	6 (5–8) [0–20]
Previous exposure to antibiotics	
Carbapenem, n (%)	219 (91)
Cephalosporin, n (%)	175 (73)
Fluoroquinolone, n (%)	100 (42)
β-lactam/β-lactamase inhibitor, n (%)	191 (80)
Aminoglycoside, n (%)	49 (20)
**Clinical characteristics**	
Initial ICU admission, n (%)	109 (45)
APACHE II score, median (IQR), [range]	17 (13–21) [9–28]
Bloodstream infection due to *S. maltophilia*, n (%)	121 (50)
Invasive medical devices, n (%)	190 (79)
Intravascular device, n (%)	123 (51)
Urinary catheterization, n (%)	187 (78)
**Treatment** Empirical treatment including	
Carbapenem(s), n (%)	234 (98)
Colistin, n (%)	73 (30)
Trimethoprim-sulfamethoxazole, n (%)	31 (13)
Fluoroquinolone(s), n (%)	18 (8)
Ciprofloxacin, n (%)	9 (4)
Levofloxacin, n (%)	9 (4)
Appropriate empirical antibiotics, n (%)	105 (44)
Colistin monotherapy, n (%)	63 (26)
Colistin plus trimethoprim-sulfamethoxazole or fluoroquinolone, n (%)	42 (18)
Duration of empirical treatment (day), median (IQR), [range]	3 (2–3) [2–4]
Definitive treatment regimen(s)	
Colistin plus trimethoprim-sulfamethoxazole, n (%)	153 (64)
Colistin plus fluoroquinolone(s), n (%)	19 (8)
Ciprofloxacin, n (%)	10 (4)
Levofloxacin, n (%)	9 (4)
Trimethoprim-sulfamethoxazole monotherapy, n (%)	37 (15)
Fluoroquinolone monotherapy, n (%)	10 (4)
Ciprofloxacin, n (%)	7 (3)
Levofloxacin, n (%)	3 (1)
Colistin monotherapy, n (%)	21 (9)
Duration of definitive treatment (day), median (IQR), [range]	14 (12–19) [7–21]

VAP, ventilator-associated pneumonia; IQR, interquartile range; ICU, intensive care unit; PACHE, acute physiology and chronic health evaluation.

**Table 3 antibiotics-13-00729-t003:** Comparison of characteristics and outcomes of patients with *S. maltophilia* VAP stratified by treatment type.

Parameter	*S. maltophilia* VAP Patients Who Did Not Receive Appropriate Empirical Antibiotics (N = 135)	*S. maltophilia* VAP Patients Who Received Appropriate Empirical Antibiotic with Colistin Monotherapy (N = 63)	*S. maltophilia* VAP Patients Who Received Appropriate Empirical Antibiotic with Colistin plus Trimethoprim-Sulfamethoxazole or Fluoroquinolone(s) (N = 42)	*p-*Value
**Demographics**				
Age, median (IQR)	45 (41–68)	45 (34–74)	47 (42–74)	0.639
Male sex, n (%)	82 (61)	46 (73)	30 (71)	0.166
Comorbidities, n (%)	117 (87)	58 (92)	42 (100)	**0.033**
Immunocompromised status, n (%)	6 (4)	3 (5)	3 (7)	0.662
Obesity, n (%)	102 (76)	41 (65)	33 (79)	0.210
Diabetes mellitus, n (%)	52 (39)	17 (27)	13 (31)	0.250
Hypertension, n (%)	70 (52)	32 (51)	22 (52)	0.899
Chronic kidney disease, n (%)	20 (15)	7 (11)	3 (7)	0.392
Cardiovascular disease, n (%)	30 (22)	9 (14)	9 (21)	0.072
Cerebrovascular disease, n (%)	4 (3)	3 (5)	1 (2)	0.791
Chronic pulmonary disease, n (%)	38 (28)	15 (24)	12 (29)	0.792
Solid organ malignancy, n (%)	4 (3)	1 (2)	1 (2)	0.134
Hematologic malignancy, n (%)	22 (16)	18 (29)	9 (21)	0.662
Charlson comorbidity index, median (IQR)	6 (4, 7)	7 (5, 9)	7 (6, 9)	0.234
**Clinical characteristics**				
Initial ICU admission, n (%)	48 (36)	37 (59)	24 (57)	**0.002**
APACHE II score, median (IQR), [range]	16 (14–18) [7−23]	21 (14–23) [7−22]	20 (13–22) [8−23]	**<0.001**
Bloodstream infection due to *S. maltophilia*, n (%)	67 (50)	32 (51)	22 (52)	0.950
Invasive medical devices, n (%)	101 (75)	53 (84)	36 (86)	0.167
Previous exposure to carbapenems, n (%)	124 (92)	58 (92)	37 (88)	0.754
**Treatment**				
Duration of empirical treatment (day), median (IQR)	3 (2, 3)	3 (2, 3)	3 (2, 3)	0.867
Empirical treatment including carbapenems, n (%)	132 (98)	61 (98)	41 (98)	0.903
Definitive treatment regimen(s)				
Colistin plus trimethoprim-sulfamethoxazole, n (%)	86 (64)	39 (62)	28 (66)	0.764
Colistin plus fluoroquinolone(s), n (%)	11 (8)	4 (6)	5 (12)	0.064
Trimethoprim-sulfamethoxazole monotherapy, n (%)	22 (16)	9 (14)	6 (14)	0.201
Fluoroquinolone monotherapy, n (%)	4 (3)	3 (5)	3 (7)	0.074
Colistin monotherapy, n (%)	12 (9)	5 (8)	4 (10)	0.815
Duration of definitive treatment (day), median (IQR)	15 (13–20)	14 (11–18)	14 (12–19)	0.564
**Outcomes**				
Mortality				
14-day, n (%)	42 (31)	13 (21)	1 (2)	**<0.001**
30-day, n (%)	60 (44)	19 (30)	2 (5)	**<0.001**
In-hospital, n (%)	71 (53)	19 (30)	5 (12)	**<0.001**
Length of hospital stay after end of VAP treatment (days), median (IQR)	28 (12–43)	29 (21–43)	21 (15–32)	**0.032**
No. of ventilator days since diagnosis of VAP (days), median (IQR)	17 (14–21)	15 (12–20)	11 (9–17)	**0.041**
Hospital cost (Baht), median (IQR)	200,716 (124,897–289,326)	188,896 (115,345–232,886)	156,564 (106,887–190,723)	**0.007**

Invasive medical devices include intravascular catheters, urinary catheters, and catheters for pus or content drainage. Boldface entries indicate values that reached the significance level set at 0.05. VAP, ventilator-associated pneumonia; IQR, interquartile range; ICU, intensive care unit; APACHE, acute physiology and chronic health evaluation.

**Table 4 antibiotics-13-00729-t004:** Factors associated with in-hospital mortality in 240 patients with *S. maltophilia* VAP.

Variables	Values		Crude ORs (95% CI)	Adjusted ORs (95% CI)	*p-*Values for Adjusted ORs
	Survivors (N = 145)	Non-Survivors (N = 95)			
Age (years), median (IQR)	44 (41–67)	48 (40–75)	1.13 (0.98, 1.38)	1.02 (0.97, 1.12)	0.099
Male sex, n (%)	102 (70)	56 (59)	0.82 (0.44, 1.54)	0.60 (0.35, 1.04)	0.548
Charlson comorbidity index, median (IQR)	7 (5, 9)	8 (6, 9)	1.23 (0.88, 1.45)	1.04 (0.81, 1.24)	0.675
Immunocompromised status, n (%)	9 (6)	3 (3)	0.63 (0.15, 2.70)	0.49 (0.12, 1.85)	0.528
Obesity, n (%)	108 (75)	68 (72)	0.86 (0.48, 1.5)	0.77 (0.38, 1.51)	0.434
APACHE II score [median (IQR)]	14 (12–18)	19 (15–22)	1.88 (1.21, 3.02)	1.32 (1.05, 2.43)	**0.041**
Bloodstream infection due to *S. maltophilia*, n (%)	81 (56)	40 (42)	0.57 (0.34, 0.97)	0.88 (0.56, 1.16)	0.091
Initial intensive care unit admission, n (%)	69 (48)	40 (42)	0.80 (0.48, 1.35)	1.41 (0.75, 2.63)	0.286
Appropriate empirical antibiotic(s)					
Colistin monotherapy, n (%)	44 (30)	19 (20)	0.39 (0.21, 0.74)	0.25 (0.12, 0.53)	**<0.001**
Colistin plus trimethoprim-sulfamethoxazole or fluoroquinolone(s), n (%)	37 (26)	5 (5)	0.12 (0.05, 0.33)	0.08 (0.02, 0.19)	**<0.001**

Boldface entries indicate values that reached the significance level set at 0.05. VAP, ventilator-associated pneumonia; IQR, interquartile range; APACHE, acute physiology and chronic health evaluation.

## Data Availability

Data are contained within this article.

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
