# Peer review of "Impact of Empirical Antimicrobial Treatment on Patients with Ventilator-Associated Pneumonia Due to Stenotrophomonas maltophilia"

_antibiotics, 2024, doi:10.3390/antibiotics13080729_

Round 1
Reviewer 1 Report
Comments and Suggestions for Authors
Dear reviewers, I had the pleasure of reading your manuscript entitled "Impact of empirical antimicrobial treatment on patients with ventilator-associated pneumonia due to Stenotrophomonas maltophilia". The study was conducted in a single tertiary hospital, and I believe that the manuscript is of interest to the clinical personnel at your hospital. However, this approach may not be applicable to the scientific community.
I have some observations
1. Table 2, 3 and 4 (IQR), [range] numbers could be separated by a hyphen instead of a comma. In addition, we added the statistical analysis and p-values to consider it significant ​​in the footer of the table.
2. What was the criterion for not including all variables in the univariate odds ratios? In addition, the study lacked multivariate odds ratios.
3. A description of the results of the odds ratio analysis is not included in the results section, also it not discussed with the results of other studies.
4. The discussion section is very poor, they should improve it
I hope that these comments will be helpful in improving the manuscript.
Author Response
Response to the reviewers
We appreciate the insightful comments from all reviewers, which have significantly enhanced the quality of our manuscript. Below are our detailed responses to each comment:
Reviewer 1:
- Table 2, 3 and 4 (IQR), [range] numbers could be separated by a hyphen instead of a comma. In addition, we added the statistical analysis and p-values to consider it significant in the footer of the table.
Response: We have updated Tables 2, 3, and 4 as per the recommendations, separating IQR [range] numbers with hyphens and including statistical analysis and p-values in the footers.
- What was the criterion for not including all variables in the univariate odds ratios? In addition, the study lacked multivariate odds ratios.
Response: We have added the criteria for including variables in multivariate analysis in the Materials and Methods (Statistical Analysis) section: "Variables with p-values < 0.2 and variables with potential clinical relevance were included in multivariate logistic regression models" (Lines 282-284).
- A description of the results of the odds ratio analysis is not included in the results section, also it not discussed with the results of other studies.
Response: We have added the odds ratio into the results section as follows: “Only the initial APACHE II score and use of appropriate empirical antibiotics, including colistin-based regimens and colistin plus TMP-SMX or fluoroquinolone-based regimens, were associated with in-hospital mortality with adjusted OR (95% CI) of 1.32 (1.05,2.43), 0.25 (0.12,0.53) and 0.08 (0.02,0.19), respectively.” in Results Lines 152-155.
We have added this perspective into the discussion as follows: “In-hospital mortality rate among the patients who did not receive appropriate empirical antimicrobial agents was relatively high (53%). This finding was like previous studies of VAP due to S. maltophilia [2,3,13,14]. In this study, APACHE II, indicating the severity of VAP due to this organism was significantly associated with in-hospital mortality with OR (95%CI) of 1.32 (1.05,2.43). Like previous studies of VAP due to several gram-negative bacteria, the initial conditions of the patients with VAP are independent of the appropriateness of empirical antimicrobial regimens [18-20]. Resemble to previous studies, appropriate empirical antimicrobial regimens for the patients with VAP due to S. maltophilia were significantly associated with lower mortality [13,14, 18-20]. Thus, our study demonstrated the different appropriate empirical regimens yielded different effects on the mortality of the patients. These findings filled the gap of previous studies on VAP due to S. maltophilia which demonstrated the benefits of appropriate empirical antimicrobial agents regardless of different types of antimicrobial agents [2,3,16,17].” In Discussions Lines 180-192
- The discussion section is very poor, they should improve it
Response: We have modified the Discussion as follows:
- We have added “In-hospital mortality rate among the patients who did not receive appropriate empirical antimicrobial agents was relatively high (53%). This finding was like previous studies of VAP due to S. maltophilia [2,3,13,14]. In this study, APACHE II, indicating the severity of VAP due to this organism was significantly associated with in-hospital mortality with OR (95%CI) of 1.32 (1.05,2.43). Like previous studies of VAP due to several gram-negative bacteria, the initial conditions of the patients with VAP were independent of the appropriateness of empirical antimicrobial regimens [18-20]. Resemble to previous studies, appropriate empirical antimicrobial regimens for the patients with VAP due to S. maltophilia were significantly associated with lower mortality [13,14, 18-20]. Thus, our study demonstrated the different appropriate empirical regimens yielded different effects on the mortality of the patients. These findings filled the gap of previous studies on VAP due to S. maltophilia which demonstrated the benefits of appropriate empirical antimicrobial agents regardless of different types of antimicrobial agents [2,3,16,17].” Lines 180-192
- We have modified the following sentences accordingly: “The current study showed the clinical and non-clinical benefits associated with appropriate antimicrobial treatment. These findings suggest that the use of appropriate empirical antibiotics yielded lower 14-day, 30-day, and in-hospital mortality rates at approximately 29%, 24%, and 30%, respectively, as well as decreases in the length of hospital stay, duration of ventilation, and hospital costs, respectively. These findings are consistent with those of studies on VAP that focused on empirical antibiotic regimens [18–20]. Additionally, this current study showed lower mortality among the patients who received appropriate empirical antibiotics with colistin plus trimethoprim-sulfamethoxazole or fluoroquinolone(s), compared to those who received empirical antibiotics with colistin monotherapy with approximately lower 14-day, 30-day, and in-hospital mortality rates of 12%, 14%, and 17%, respectively.” Lines 194-204
- We have modified the following sentence accordingly: “Fourth, the data used in this study were retrieved from January 2010 to December 2021, and there are no standard recommendations for the definitive treatment of infections caused by maltophilia.” Lines 221-223
- We have added, "Last, the definition of appropriate empirical antimicrobial agents was determined by only antimicrobial susceptibility testing regardless of pharmacokinetic properties as well as the dosage of antimicrobial agents which had potential impacts to outcomes of the patients.” Lines 226-229
Sincerely yours,
Sarunyou Chusri
Corresponding authors
Reviewer 2 Report
Comments and Suggestions for Authors
How the appropriate antibiotics was decided should be properly justified rather than deciding on ABST results.
91%of the sample had prior carbapenem treatment, have you considered the effect of this when analyzing?
As a limitation, the authors have mentioned that the data was retrieved in Jan 2010 and in another place it says data collection duration was Jan 2010 to Dec 2021, have you collected all the data for 11 years or collected data at 2 time points need to be clarified.
Author Response
Response to the reviewers
We appreciate the insightful comments from all reviewers, which have significantly enhanced the quality of our manuscript. Below are our detailed responses to each comment:
Reviewer 2:
- How the appropriate antibiotics was decided should be properly justified rather than deciding on ABST results.
Response: We agree that the appropriate empirical antimicrobial agents with the definition of this current study might not be appropriate to the patients then we clearly emphasized this definition in part of Methods as follows: “Appropriate empirical antimicrobial agents were determined using only susceptibility testing information without concerning to pharmacokinetic properties and administration dosages.” Lines 269-271 and we also mentioned this limitation in part of the Discussion as follows: “Last, the definition of appropriate empirical antimicrobial agents was determined by only antimicrobial susceptibility testing regardless to pharmacokinetic properties as well as the dosage of antimicrobial agents which had potential impacts to outcomes of the patients.” Lines 226-229.
- 91%of the sample had prior carbapenem treatment, have you considered the effect of this when analyzing?
Response: We clarified the sentence: “All isolates were resisted to meropenem and imipenem with MIC50 of 128 and 128 μg/mL, respectively, and MIC90 512 and 512 μg/mL, respectively.” In Results Lines 89-91 and previous exposure to carbapenems and empirical treatment including carbapenems in Table 3. However, we did not put empirical treatment including carbapenems into multivariate analysis for 2 reasons; first, the rate of empirical treatment including carbapenem among the three groups was not different (P-value of 0.897), and the relatively high MIC which was mentioned in Lines 89-91.
- As a limitation, the authors have mentioned that the data was retrieved in Jan 2010 and in another place, it says the data collection duration was Jan 2010 to Dec 2021, have you collected all the data for 11 years or collected data at 2 time points need to be clarified.
Response: We apologize for the unclear sentences. We have modified the sentence as follows: “Fourth, the data used in this study were retrieved from January 2010 to December 2021, and there are no standard recommendations for the definitive treatment of infections caused by
- S. maltophilia.” In Discussions Lines 221-223.
Sincerely yours,
Sarunyou Chusri
Corresponding authors
Reviewer 3 Report
Comments and Suggestions for Authors
The significance of antimicrobial treatment on patients is very important globally. The current study suggests that the use of appropriate empirical antimicrobial treatment based on spectra and pharmacokinetics improves the survival of patients with VAP due to S. maltophilia and decreases hospitalization rates for this infection.
The title is informative, the aim is clear. However, the introduction of this article does not successfully explain why the understanding this problem is very important.
Here are some suggestions that would improve the manuscript:
1.According to the context in the Introduction, the authors can give a
a little more information about: S. maltophilia, about the spread, about the horizontal transfer of antibiotic resistance, also about on a worldwide - what is happening, because the S.maltophilia is globally distributed and is in the group of microorganisms whose resistance is of primary importance for public health, and considering clinical isolates and environmental isolates - is there any information on which have a higher mutation rate and association with the virulence
2. line 43-47 - Which are inappropriate antimicrobial treatment options?- give them or why are inappropriate
The “Results” - clearly presented and illustrated with tables.What about the mortality patients? Give the numbers in the text in the results, because it is difficult to follow them in the table,graphs and text.
The “Discussion” - The discussion volume is quite large and the problem is discussed from different angles but again the practical meaning of them is not clear enough.
The “Conclusion” section, on the other hand, can be improved. It can be more convincing. It is necessary to explain how this research can be used in practice.
Chapter “Materials and Methods” - the methods adequately described, valid and reliable. The variables are well defined and measured appropriately.
Minor editing
Author Response
Response to the reviewers
We appreciate the insightful comments from all reviewers, which have significantly enhanced the quality of our manuscript. Below are our detailed responses to each comment:
Reviewer 3:
- According to the context in the Introduction, the authors can give a little more information about: maltophilia, about the spread, about the horizontal transfer of antibiotic resistance, also about on a worldwide - what is happening, because the S.maltophilia is globally distributed and is in the group of microorganisms whose resistance is of primary importance for public health, and considering clinical isolates and environmental isolates - is there any information on which have a higher mutation rate and association with the virulence
Response: We have added the following sentences in the Introductions: “Thus, this organism has been found as a contaminant in hospital environments, particularly in several kinds of medical devices, specimen collection tubes, chlorhexidine-based disinfectant, drinking water supply, and chlorinated sterile water. [3]. In addition, S. maltophilia exhibits intrinsic resistance to numerous antimicrobial agents with intrinsic mechanisms including multidrug-efflux pumps and low permeability as well as the acquisition of resistance through the uptake of resistance genes located on integrons, transposons, and plasmids which are responsible for β-lactamases, aminoglycoside-modifying enzymes and alteration of folate synthesis [1, 3]. Infections due to S. maltophilia have raised concern with the number of surveillance cases extremely increasing, particularly in European and Southeast Asian& Pacific countries [1,3,8,9]. These infections commonly emerged among vulnerable patients with severely compromised health status, retaining medical, exposure to broad-spectrum antimicrobials, and long hospital stays. [1,3] Occurrence of outbreaks with local spreading of these infections were frequently noticeable as well as global spreading of specific strains were identified [2,3,8,9].” Lines 44-58
- Line 43-47 - Which are inappropriate antimicrobial treatment options? - give them or why are inappropriate
Response: We apologize for the unclear sentences. We have rewritten the following sentences in part of the Introductions: “Although S. maltophilia is classified as a non-inherently virulent organism, VAP caused by this organism has a high mortality rate owing to the limitation of appropriate antimicrobial treatment options [1,8,9]. It can be explained as follows: First,
carbapenems which were usually used as empirical treatment for hospital-acquired infection, had unfavorable antimicrobial susceptibility due to intrinsic resistance of this organism [9], second, although S. maltophilia is susceptible to colistin, commonly empirically prescribed for suspected carbapenem-resistant gram-negative bacterial infections, the pulmonary tissue penetration ability of this agent is relatively poor [10,11], third, even cotrimoxazole (trimethoprim-sulfamethoxazole [TMP-SMX]) and fluoroquinolones which has high susceptibility against S. maltophilia, these antimicrobial agents are not commonly prescribed for hospital-acquired infection due to poor susceptibility carbapenem-resistant Enterobacterales (CRE), carbapenem-resistant Acinetobacter baumannii (CRAB) and difficult-to-treat resistant Pseudomonas aeruginosa (DTRPA). [12] Then, treatment of VAP caused by S. maltophilia is extremely hampered because this organism is intrinsically resistant to several kinds of antimicrobial agents thus the existing potential antimicrobial agents against this organism have unclear values for in vitro susceptibility testing [8,9].” Lines 59-75
- The “Results” - clearly presented and illustrated with tables. What about the mortality patients? Give the numbers in the text in the results, because it is difficult to follow them in the table, graphs and text.
Response: We have modified the following sentences: “The 14-day mortalities of the patients who did not receive appropriate empirical antimicrobial agents (31%) was significantly different from those received empirical antimicrobial treatment with colistin monotherapy-based regimen (21%) and those received empirical antimicrobial treatment with colistin plus TMP-SMX or fluoroquinolone-based regimen (2%) (P-value < 0.001). The 30-day mortalities of the patients who did not receive appropriate empirical antimicrobial agents (44%) were significantly different from those who received empirical antimicrobial treatment with colistin monotherapy-based regimen (30%) and those who received empirical antimicrobial treatment with colistin plus TMP-SMX or fluoroquinolone-based regimen (5%) (P-value < 0.001). The in-hospital mortalities of the patients who did not receive appropriate empirical antimicrobial agents (53%) were significantly different from those who received empirical antimicrobial treatment with colistin monotherapy-based regimen (30%) and those who received empirical antimicrobial treatment with colistin plus TMP-SMX or fluoroquinolone-based regimen (12%) (P-value < 0.001). The length of hospital stay after VAP treatment, duration of ventilation since VAP diagnosis, and hospital cost, among those who received appropriate empirical antimicrobial treatment, were significantly favorable, particularly in those who received colistin plus TMP-SMX or fluoroquinolone-based regimen, compared to those who did not receive appropriate empirical antimicrobial agents.” In Results Lines 120-138.
- The “Discussion” - The discussion volume is quite large and the problem is discussed from different angles but again the practical meaning of them is not clear enough.
Response: We have modified the Discussion as follows:
- We have added “In-hospital mortality rate among the patients who did not receive appropriate empirical antimicrobial agents was relatively high (53%). This finding was like previous studies of VAP due to S. maltophilia [2,3,13,14]. In this study, APACHE II, indicating the severity of VAP due to this organism was significantly associated with in-hospital mortality with OR (95%CI) of 1.32 (1.05,2.43). Like previous studies of VAP due to several gram-negative bacteria, the initial conditions of the patients with VAP were independent of the appropriateness of empirical antimicrobial regimens [18-20]. Resemble to previous studies, appropriate empirical antimicrobial regimens for the patients with VAP due to S. maltophilia were significantly associated with lower mortality [13,14, 18-20]. Thus, our study demonstrated the different appropriate empirical regimens yielded different effects on the mortality of the patients. These findings filled the gap of previous studies on VAP due to S. maltophilia which demonstrated the benefits of appropriate empirical antimicrobial agents regardless of different types of antimicrobial agents [2,3,16,17].” Lines 180-192
- We have modified sentences “The current study showed the clinical and non-clinical benefits associated with appropriate antimicrobial treatment. These findings suggest that the use of appropriate empirical antibiotics yielded lower 14-day, 30-day, and in-hospital mortality rates at approximately 29%, 24%, and 30%, respectively, as well as decreases in the length of hospital stay, duration of ventilation, and hospital costs, respectively. These findings are consistent with those of studies on VAP that focused on empirical antibiotic regimens [18–20]. Additionally, this current study showed lower mortality among the patients who received appropriate empirical antibiotics with colistin plus trimethoprim-sulfamethoxazole or fluoroquinolone(s), compared to those who received empirical antibiotics with colistin monotherapy with approximately lower 14-day, 30-day, and in-hospital mortality rates of 12%, 14%, and 17%, respectively.” Lines 194-204
- We have modified the following sentence accordingly: “Fourth, the data used in this study were retrieved from January 2010 to December 2021, and there are no standard recommendations for the definitive treatment of infections caused by maltophilia.” Lines 221-223
- We have added, "Last, the definition of appropriate empirical antimicrobial agents was determined by only antimicrobial susceptibility testing regardless of pharmacokinetic properties as well as the dosage of antimicrobial agents which had potential impacts to outcomes of the patients.” Lines 226-229
- The “Conclusion” section, on the other hand, can be improved. It can be more convincing. It is necessary to explain how this research can be used in practice.
Response: We have modified the Conclusions as follows: “VAP due to S. maltophilia yielded a high mortality rate and excessive hospital expenditures. Appropriate empirical antimicrobial treatment in terms of both susceptibility and pharmacokinetics tended to improve clinical outcomes and decrease ventilator days, length of hospital stays, and hospital expenditures. These empirical antimicrobial agents should be considered for treating VAP among patients with medical comorbidities and prior carbapenem exposure. Well-designed clinical trials should be conducted to determine the impact of appropriate empirical antibiotic regimens on outcomes for patients at risk of VAP due to S. maltophilia.” Lines 294-301
Sincerely yours,
Sarunyou Chusri
Corresponding author
Round 2
Reviewer 1 Report
Comments and Suggestions for Authors
Dear reviewers, the manuscript was improved; however, the study was conducted in a single tertiary hospital, and I believe that the manuscript is of interest to clinical personnel at your hospital. What would be of interest to the scientific community or to people from other hospitals?
Why the paragraph "All isolates were resistant to meropenem and imipenem with MIC50 values of 128 and 128 μg/mL, respectively, and MIC90 512 and 512 μg/mL, respectively”, could be important when S. maltophilia is intrinsically resistant to carbapenems?
In addition, how did you calculate MIC50 and MIC90?
In tables 2, 3 and 4 it is not indicated that the number in parentheses is %
Author Response
Response to reviewer 1
Dear reviewers, the manuscript was improved; however, the study was conducted in a single tertiary hospital, and I believe that the manuscript is of interest to clinical personnel at your hospital. What would be of interest to the scientific community or to people from other hospitals?
Response: Thank you for your thorough review of our manuscript. We have added the following sentence to address your concern: “First, this study was conducted in a single tertiary hospital, and the findings can be generalized only to similar settings. Understanding that the number of patients with VAP due to S. maltophilia in other settings is relatively low, this study demonstrated the clinical characteristics of patients with VAP due to S. maltophilia, suggesting the use of empirical antimicrobial agents.” Lines 208-212
Why the paragraph "All isolates were resistant to meropenem and imipenem with MIC50 values of 128 and 128 μg/mL, respectively, and MIC90 512 and 512 μg/mL, respectively”, could be important when S. maltophilia is intrinsically resistant to carbapenems?
In addition, how did you calculate MIC50 and MIC90?
Response: We apologized for miscommunication from previous discussion. We only showed the relatively high MIC values of carbapenem for S. maltophilia that might not be affected by patient outcomes. We have deleted the mentioned sentences from the Results section.
In tables 2, 3 and 4 it is not indicated that the number in parentheses is %
Response: Thank you for pointing this out. We have added “(%)” in all tables.